# CAG*n* Polymorphic Locus of Androgen Receptor (*AR*) Gene in Russian Infertile and Fertile Men

**DOI:** 10.3390/ijms252212183

**Published:** 2024-11-13

**Authors:** Vyacheslav Chernykh, Olga Solovova, Tatyana Sorokina, Maria Shtaut, Anna Sedova, Elena Bliznetz, Olga Ismagilova, Tatiana Beskorovainaya, Olga Shchagina, Aleksandr Polyakov

**Affiliations:** 1Research Centre for Medical Genetics, 115522 Moscow, Russia; olga_pilyaeva@list.ru (O.S.); reprolab@med-gen.ru (T.S.); shtaut@yandex.ru (M.S.); luoravetlanka@gmail.com (A.S.); bliznetzelena@mail.ru (E.B.); ismolga.mg@gmail.com (O.I.); t-kovalevskaya@yandex.ru (T.B.); schagina@med-gen.ru (O.S.); polyakov@med-gen.ru (A.P.); 2Pirogov Russian National Research Medical University of the Ministry of Healthcare of the Russian Federation, 117997 Moscow, Russia

**Keywords:** androgen receptor, male infertility, spermatogenesis, pathozoospermia, fertility, trinucleotide repeats, spinal and bulbar muscular atrophy, SBMA (Kennedy disease), X chromosome

## Abstract

The androgen receptor (AR) is critical for mediating the effects of androgens. The polymorphic CAG*n* locus in exon 1 of the *AR* gene is associated with several diseases, including spinal and bulbar muscular atrophy (SBMA), prostate cancer, and male infertility. This study evaluated the CAG*n* locus in 9000 infertile Russian men and 286 fertile men (control group). The CAG*n* locus was analyzed using the amplified fragment length polymorphism method. In the infertile cohort, the number of CAG repeats ranged from 6 to 46, with a unimodal distribution. The number of CAG repeats in infertile and fertile men was 22.15 ± 0.93 and 22.02 ± 1.36, respectively. In infertile men, variants with 16 to 29 repeats were present in 97% of the alleles. A complete mutation (≥42 CAG repeats) was found in three patients, while three others had 39-41 repeats. The incidence of SBMA was 1:3000 infertile men. Significant differences (*p* < 0.05) were observed between infertile and fertile men in alleles with 21, 24 and 25 repeats. This study revealed certain differences in the CAG*n* polymorphic locus of the *AR* gene in Russian infertile and fertile men and determined the frequency of SBMA in infertile patients.

## 1. Introduction

Infertility is one of the most common medical and social problems of our time. Male infertility and subfertility are usually associated with abnormal basic semen parameters (sperm concentration, motility, vitality and morphology), which are detected in 50-60% of all men from infertile couples [1]. Various pathogenic factors affecting the reproductive system, including congenital anomalies and genetic defects, hormonal, oncological and multifactorial disorders, infections, toxins and environmental pollutants can negatively affect spermatogenesis, testis, epididymis, seminal ducts and other male reproductive organs, reducing semen parameters and leading to male infertility [2]. 

Androgens are male sex hormones that play a pivotal role in the endocrine regulation of the development and function of the male reproductive system and skeletal growth, metabolism and homeostasis in both mammals and humans by acting on target genes through the activation of the androgen receptor (AR) protein, which regulates the functions of all androgen-dependent processes [3]. AR is a ligand-dependent transcription factor that binds to androgens and is required for the development of the male reproductive organs and the hormonal regulation of fertility, the control of mitotic and meiotic divisions of immature male germ cells, affecting spermiogenesis, the maturation of spermatozoa in the epididymis, male accessory sex glands, and other biological functions and processes [4]. 

In humans, the androgen receptor (*AR*; OMIM: *313700) gene, located on the X chromosome (Xq12 locus), is 90 Kb in size and contains 8 exons [5]. The protein encoded by the *AR* gene contains three major domains: the N-terminal transactivating domain (encoded by exon 1), the DNA-binding domain (encoded by exons 2 and 3) and the ligand-binding domain—LBD (encoded by exons 4–8) [6]. Exon 1 contains two polymorphic trinucleotide repeat sites (CAG and CGG) that encode the polyglutamine and polyglycine tracts, respectively, in the AR protein. Normally, the length of the CAG*n* polymorphic locus of the *AR* gene varies from 7 to 37 trinucleotide repeats, and the CGG repeat length is usually between 10 and 36 [7]. 

Pathogenic variants in the *AR* gene result in structural and functional defects of the AR protein, leading to various genetic disorders, including complete (testicular feminization) and varying degrees of incomplete insensitivity to androgens (androgen insensitivity syndrome, AIS) and male infertility associated with azoospermia or oligozoospermia [8,9]. Somatic pathogenic variants of the *AR* gene have been described in both patients with androgen insensitivity syndrome and prostate cancer [10]. Mild defects in AR function are one of the genetic causes of impaired spermatogenesis and male infertility with no undermasculinization [11]. Polymorphic trinucleotide variants can affect androgen receptor (AR) function, the alleles with a smaller number of CAG repeats are characterized by an increased transactivation function of AR protein and androgen sensitivity and those with a larger number of repeats are characterized by decreased androgen sensitivity [12]. An increase in the number of CAG repeats (more than 38–40) in exon 1 of the *AR* gene leads to the development of a progressive X-linked recessive neuromuscular disease—spinal and bulbar muscular atrophy, SBMA (OMIM #313300), also known as Kennedy disease [13]. Due to hemizygosity on the X chromosome, Kennedy disease is much more common in males than in females. SBMA is characterized by a late onset, with clinical manifestations of the disease usually occurring in patients over the age of 30–40 years. In rare cases with extreme increases in the number of CAG repeats, the onset of the disease is noted in younger patients [14]. 

Numerous studies have shown an association between the *AR* CAG polymorphic locus and male subfertility or infertility in different populations and ethnic groups [15,16,17,18]; however, some researchers have failed to demonstrate this association [19]. Some variations in the CAG repeats in exon 1 of the *AR* gene have been observed in different populations, as well as in fertile and infertile men from different regions. For example, African Americans generally have fewer repeats than non-Hispanic whites [20].

Previously, the CAG*n* polymorphic locus of the *AR* gene has been studied in fertile and infertile men from the Russian Federation and Ukraine, and in healthy Russian men of different ethnic groups (Slavic, Buryat and Yakut) [21,22,23,24]. The *AR* gene variant with 21 CAG repeats is the most common in Russian and Ukrainian infertile men and healthy Slavic men. However, these studies involved much smaller sample sizes and did not calculate or compare the frequency of individual CAG allelic variants between the examined groups and other cohorts. Although many authors indicate an increased frequency of “long” CAG repeats, the frequency of complete expansion of trinucleotide repeats in the *AR* gene, also known as Kennedy syndrome, in infertile men has not been evaluated. 

The aim of the study is to evaluate the CAG*n* polymorphic locus in exon 1 of the *AR* gene and to determine the frequency of its different allelic variants in a large cohort of infertile Russian men and to compare them with fertile Russian men. 

## 2. Results

None of the examined individuals in both groups were heterozygous for the CAG*n* polymorphic locus in exon 1 of the *AR* gene. In the examined cohort (infertile men), the number of CAG repeats varied from 6 to 46 (Figure 1). 

In fertile men (controls), the number of CAG repeats ranged from 13 to 31 (Figure 2). In the group of infertile patients, the median number of CAG repeats was 22, and in the group of fertile men the median was 21 (Table 1).

The distribution of the different CAG alleles of the *AR* gene was unimodal in infertile and near to unimodal in fertile individuals, very similar to the symmetric distribution of CAG repeats in infertile patients. In both the infertile men and controls, the most common allele (Moda) contains 21 trinucleotide repeats, Q25 and Q75 quantiles consist of 20 and 24 repeats. The mean number of CAG repeats in infertile and fertile men was very similar and consisted of 22.2 ± 3.1 and 22.0 ± 2.7, respectively (Table 1). 

Common *AR* gene variants with 16 to 29 CAG repeats were found with allelic frequencies (AF) ranging from 0.0110 to 0.1698. In total, these variants represent 97.04% and 1.75% of all alleles detected in the cohort of infertile and fertile Russian men, respectively. Allelic variants with an AF greater than 0.01 (1% of all alleles) are shown in Table 2.

In both Russian infertile men and fertile individuals (controls), *AR* gene variants with 20 to 24 CAG repeats were more common, with a frequency of more than 10% each in the studied sample (Table 2).

Alleles with the number of CAG repeats between 6 and 15 and 30 and 46 were found much less frequently. Their proportion among all detected allelic variants was 2.96% (Figure 1 and Figure 2). Each of the variants with the number of CAG repeats *n* ≥ 39 was detected once; in total, they were found in six (0.07%) patients (Figure 1). In fertile men (control group), the largest number of repeats was 31, detected in only one (0.35%) of 286 individuals (Figure 2). 

A complete mutation (CAG repeats, *n* ≥ 42) in exon 1 of the *AR* gene, characteristic of spinal and bulbar muscular atrophy (SBMA), was found in three patients (all of whom had developed SBMA at the time of examination); and in three infertile men the number of repeats was *n* = 39–41 (with no diagnosed SBMA at the time of examination) (Figure 1). Thus, the incidence of SBMA in the sample studied was estimated to be 1 per 3000 infertile Russian men or 1 per 1500 infertile patients if carriers of alleles with 39–41 CAG repeats are considered.

A comparative analysis of each of the common CAG*n* alleles in infertile men and controls revealed statistically significant (*p* < 0.05) differences for three polymorphic variants of the *AR* gene containing 21, 24 and 25 trinucleotide repeats. The allelic frequency (AF) of the *AR* gene variants with 21 and 25 CAG repeats was lower in infertile men and higher in the controls (AF 0.1698 vs. 0.2273; *p* = 0.012, and 0.0744 vs. 0.1189; *p* = 0.006, respectively).

The allele with 24 CAG repeats was more frequent in infertile Russian men than in fertile individuals (0.1142 vs. 0.0734; *p* = 0.032) (Table 2). Statistically significant differences were found in the frequency of some common alleles (with 21, 24 and 25 CAG repeats) between infertile patients and the control group.

“Short” (*n* ≤ 18), “medium” (*n* = 19–25) and “long” (*n* ≥ 26) allelic variants of the CAG*n* polymorphic locus were detected in 722 (8.02%), 7186 (79.84%) and 1092 (12.13%) infertile patients, respectively. In fertile men (control group), “short”, “medium” and “long” alleles were found in 15 (5.24%), 247 (86.36%) and 24 (8.39%) individuals, respectively. There was a statistically significant difference in the frequency of “medium” CAG repeats and “short” and “long” repeats together (*χ*^2^ = 7.375; *p* = 0.007), but not in the groups of “short” (*χ*^2^ = 2.927; *p* = 0.088) and “long” (*χ*^2^ = 3.670; *p* = 0.056) repeats separately. 

## 3. Discussion

The CAG*n* polymorphic locus of the *AR* gene has been studied in infertile men from different populations and countries, but the size of the samples studied has been significantly smaller [15,16,17,18,19,20,21,22,23,24,25,26,27]. Many authors have shown that “long” CAG repeats of the *AR* gene increase the risk of impaired spermatogenesis and male infertility in different populations and regions. Studies performed on samples from subfertile and fertile men from different populations (European, Asian, American, African and mixed-origin populations) have shown an association of “long” (*n* ≥ 26) repeats with male infertility and reduced semen quality [7,15,16,17,20,21,22,24,25,26,27]. 

In one of the largest studied cohorts of men (*n* = 1977) from the general population of several European countries (Italy, Belgium, Poland, Sweden, United Kingdom, Spain, Hungary and Estonia), the number of CAG repeats of the *AR* gene varied between 6 and 39, with the variant with 21 CAG repeats showing the highest proportion (16.4%) of all detected alleles [28]. The distribution of CAG repeats in this cohort and our sample is very similar, especially for the common alleles (CAG*n*, *n* = 16–29).

In Russians and Ukrainians, as well as in many individuals from other populations, variants with 20 or 21 CAG repeats in exon 1 are the most common *AR* gene alleles [21,22,23]. In our cohort of infertile Russian men, these allelic variants accounted for approximately 29% of all detected *AR* gene alleles in this group. In the cohort of Russian infertile men (*n* = 332) studied by Mikhaylenko et al. (2019), the number of CAG trinucleotides varied from 9 to 31, the frequency of the *n* = 21 allele was 21.5%; “long” alleles (*n* ≥ 27 repeats) were identified in 7.5% of patients [21]. Patients with chromosomal abnormalities, in particular with Klinefelter syndrome, were not excluded from the sample of infertile men, and fertile men were not studied as a control group. The authors did not provide information on the ethnicity of the patients studied. In our sample, the proportion of such allelic variants was very close, amounting to 7.22%, with the allele containing 21 repeats being the most common.

In the examined samples, the frequencies of “short” (*n* ≤ 18) and “long” (*n* ≥ 26) CAG alleles were higher in infertile patients than in fertile men (8.02% vs. 5.24% and 12.13% vs. 8.39%, respectively), but there was no statistically significant difference in the frequency of these types of alleles between the groups separately, but a statistically significant difference (χ^2^ = 7.375; *p* = 0.007) was found in the overall frequency of “short” and “long” CAG alleles between the infertile men and controls. Perhaps if the control group had been more numerous, a statistically significant difference would have been revealed separately for “short” (*n* ≤ 18) and “long” alleles.

Previously we have evaluated the CAG*n* polymorphism of the *AR* gene in Russian patients with various forms of pathozoospermia (*n* = 591), as well as in fertile (*n* = 286) and normozoospermic (*n* = 131) men [24]. There were no significant differences in the frequencies of “short” (*n* ≤ 18), “medium” (*n* = 19–25) and “long” (*n* ≥ 26) alleles between groups of patients with different spermatological diagnoses and fertility statuses. Statistically significant (*p* < 0.01) differences were found between severe oligozoospermic patients and the controls in the frequencies of “long” (*n* ≥ 26) and “short” (*n* ≤ 18) alleles, and the CAG*n* = 25 allele between azoospermic (18.1%) and severe oligozoospermic (2.6%) patients [24]. 

Fesai et al. (2009) analyzed the CAG*n* polymorphic locus of the *AR* gene in Ukrainian infertile (*n* = 228) and fertile (*n* = 124) men [22]. The frequency of alleles with CAG repeats ≤ 18 was significantly higher (*p* < 0.01) in azoospermic (17.7%) and oligozoospermic (12.5%) patients compared to fertile individuals (control group) (2.4%). The frequency of alleles with CAG repeats *n* ≥ 28 was significantly (*p* < 0.01) higher in oligozoospermic patients (12.5%) compared to the control group (2.4%). Apparently, “short” and “long” alleles are factors associated with impaired spermatogenesis and risk of oligozoospermia in men from Russian and Ukrainian populations [24].

Recently, the CAG*n* polymorphic locus of the *AR* was evaluated in a cohort of 1324 young Russian men (median age: 23.0 years) of different ethnicities (Slavs, *n* = 697; Buryats, *n* = 208; Yakuts, *n* = 134) from the general population [23]. It should be noted that all Russians in the study, including individuals from the Slavic group, lived in the Siberian region, and there was no information about their fertility. In the Slavic, Buryat and Yakut subgroups, the most frequent CAG*n* alleles were *n* = 21 (16.3%), *n* = 22 (12.9%) and *n* = 25 (21.6%), respectively. The most frequent CAG*n* allele in the Slavic subgroup was also the same as in Russians from our sample, *n* = 21, with a very similar frequency. The number of trinucleotide repeats differed significantly between all ethnic groups (*p* < 0.001), the shortest being found in the Slavic subgroup (mean: 23.0 ± 3.1, median: 23, and range: 19–29), the longest in the Yakut subgroup (mean: 25.0 ± 2.7, median: 25, range: 21–31) and the Buryat subgroup being in the middle (mean: 24.0 ± 3.5, median: 24, range: 19–30) (Table 1). A statistically significant difference (*p* < 0.05) was found in the frequency and mean CAG repeats between groups with the normal (23.2 ± 3.3) and impaired semen quality (23.9 ± 3.2) groups. The long CAG alleles were associated with impaired semen quality in the Slavic and Buryat groups, but not in the Yakut group [23].

The number of the CAG*n* repeats in exon 1 of the *AR* gene affects the function of the androgen receptor. A critical range of 16-29 triplets is required for maximum interaction between the transactivating and the hormone-binding domains [15]. *In vitro* and *in vivo* studies have shown that increasing the length of the polyglutamine tract of the AR protein results in a linear decrease in the transcriptional capacity of the androgen receptor in mammalian cell lines [12,29,30]. However, this effect has been shown to be specific to certain cell (or tissue) types [31]. This appears to be due to the different pattern of AR regulatory proteins [32]. For example, while androgen receptor-mediated mRNA levels in prostate cells appear to be inversely proportional to the length of the polyglutamine tract [33], the opposite is observed in myoblasts [34].

Spinal and bulbar muscular atrophy (SBMA) or Kennedy disease is a rare, progressive, adult-onset, X-linked recessive disease associated with repeat extension disorders, REDs [1,2]. The pathogenic mechanism of the disease is largely unknown, but it appears that the expansion of CAG repeats results in both loss of AR function and toxic gain of function [35,36]. In male patients, SBMA progresses slowly over decades with bulbar and lower motor neuron loss, disabling muscle denervation and direct skeletal muscle involvement, leading to progressive muscle loss with weakness, fasciculations and spasticity [37]. Bulbar muscle weakness follows, leading to dysarthria and dysphagia. Effects on the male reproductive organs and endocrine systems include an increase in the level of luteinizing hormone (LH) and consequently testosterone, with signs of hypogonadism, testicular hypoplasia, decreased libido, erectile dysfunction, gynecomastia, male infertility associated with azoospermia or oligozoospermia. The average age of onset of the disease in male carriers (CAG) of *n* = 35–46 alleles varies from 44 to 68 years, while in some of them, SBMA developed between the ages of 70 and 80 years. In most patients with CAG ≥ 47 alleles, the age of manifestation ranged from 25 to 43 years [38]. 

In our study, a prominent expansion of CAG repeats (39–46) was found in six patients, including three individuals with a complete mutation (≥42 repeats), which is characteristic for spinobulbar muscular atrophy (SBMA). Thus, the frequency of SBMA in the studied cohort of Russian infertile men was estimated to be 1 per 3000 infertile Russian men. However, when the carriage of alleles with 39–41 repeats is considered, the frequency of carrying alleles that can lead to the development of the disease is twice as high and is estimated to be 1 in 1500 patients. The prevalence of the disease has been calculated to be 2.6:100,000 (1:38, 462) men in the general population [3]. A relatively high frequency of expanded SBMA-associated alleles has been found in the general population with CAG*n*, *n* ≥ 35 present in 107/100,000 individuals and CAG*n*, *n* ≥ 38 present in 27/100,000 individuals [3]. 

Recently, the frequency of CAG repeat expansion was found to be 1:3, 182, which is 10 times higher than the reported disease prevalence. Based on these data, the authors concluded that the prevalence of SBMA in the general population is underestimated and estimated it to be 1:6887 men [4]. In our sample of infertile men, the frequency of CAG*n* alleles, *n* > 38, was 4.6 times higher. Probably, it is due to that infertile men have a higher frequency of “long” CAG repeats, especially alleles associated with SBMA, than individuals from the general population. As SBMA is usually characterized by late manifestation, this leads to an underestimation of its prevalence in young patients. It also should be noted that the previously studied samples of infertile men are much smaller, and since the frequency of risk alleles is low, these samples may not have been included. In the samples of infertile men studied by other authors, the frequency of the complete expansion of CAG repeats and the frequency of SBMA were not investigated. Apparently, this is due to the small size of the samples studied (mostly not more than 1000 individuals) and the low incidence of spinal and bulbar muscular atrophy in infertile men. The study of a very large sample of infertile men made it possible to detect a small number of individuals with Kennedy disease and to determine the frequency of carriers of the expansion of the androgen receptor repeats in Russian men.

Our study has some limitations. The number of individuals in the control group was relatively small. Obviously, this did not allow us to fully assess the frequency of rare CAG*n* alleles and could have some effect on the allele frequency in fertile men. As we did not have precise information on the ethnicity of all the participants, nor on the semen parameters, it was not possible to distinguish a group of Russians and Russians from individuals of other nationalities, nor to define their spermogram. As male fertility is not the same as normozoospermia and *vice versa*, men from infertile couples partially overlap with fertile men in terms of semen parameters. Neurological examination and follow-up of patients in the dynamic were not performed. This is relevant for individuals with long CAG*n* alleles (*n* > 35), as they have a high risk of developing SMBA [3]. Therefore, the exact frequency of spinal bulbar muscular atrophy has not been determined. Further studies are needed to more precisely determine the frequency of CAG alleles in different populations of Russian Federation.

The study of the CAG polymorphic locus of the *AR* gene is not mandatory for the genetic evaluation of infertile male and female patients. This molecular genetic test is used for clinical purposes to genetically confirm the diagnosis of spinal and bulbar muscular atrophy, and to identify/exclude trinucleotide repeat expansion in male and female relatives of SBMA patients, as well as to perform the X chromosome inactivation assay analyzing the *AR* gene in heterozygous individuals. To the best of our knowledge, there are currently no clinical recommendations and no generally accepted point of view for using this genetic test to evaluate infertile patients. Because this test allows us to identify CAG polymorphic locus heterozygosity, it can be used to detect Klinefelter syndrome or its variants, X chromosome polysomies, XX-sex reversal and some variants of sex-chromosome mosaicism and chimeras (in heterozygosity). In addition, the evaluation of this polymorphic locus of the *AR* gene allows us to reveal the CAG-polymorphic risk alleles for SBMA and prostate cancer in asymptomatic and affected patients, which is particularly relevant for patients with idiopathic non-obstructive azoospermia and oligozoospermia.

## 4. Materials and Methods

### 4.1. Examined Cohort 

The cohort studied consisted of 9000 infertile Russian men. The patients were examined in 2003–2023 as part of an ongoing large-scale genetic study of male infertility at the Research Centre for Medical Genetics, RCMG (Moscow, Russia). 

Inclusion criteria were marital and reproductive age, infertility and abnormal semen parameters (at least in two semen analyses). Most of the examined infertile patients had azoospermia and oligozoospermia. Patients with Klinefelter syndrome, X chromosome polysomy, 46,XX testicular disorder of sexual development (DSD), 46,XX ovotesticular disorder of sexual development (DSD) and patients with the 46,XX/46,XY karyotype were not included in the study. For the control group, data were obtained on the frequency of different allelic variants of the *AR* gene in a group of 286 unrelated Russian men with proven fertility who had been previously studied [24]. They had at least one child and their paternity was confirmed by the results of DNA genotyping. All examined subjects were men of reproductive age, and a most of individuals in both groups were Russians and ethnic Slavs living in Moscow and the Moscow region. 

The study was approved by the Ethics Committee at the Research Centre for Medical Genetics, RCMG (Moscow, Russia). Written voluntary informed consent was obtained from each individual enrolled in the study.

### 4.2. Semen Analysis

Standard semen examination was performed by the manual assessment of semen volume (mL), sperm concentration (×10^6^/mL), motility (progressive, non-progressive motile and immotile sperm), normal abnormal morphology (percentage) manually using Nikon Eclipse E200 and Nikon Eclipse Ci (Nikon Inc., Tokyo, Japan) and Axioplan 2 microscopes (Carl Zeiss, Oberkochen, Germany) with 40× and 100× objectives according to the WHO laboratory manual for the examination and processing of human semen, WHO, 2010 [39].

### 4.3. Chromosome Analysis

Chromosome analysis was performed on FGA-stimulated cultured peripheral blood lymphocytes according to standard protocol using GTG staining, allowing us to visualize chromosome G-banding using the Axioplan 2 microscope (Carl Zeiss, Oberkochen, Germany) with 40× and 100× objectives [40]. For each sample, 20 to 30 metaphase plates were analyzed. The karyotype formula was recorded according to the International System for Human Cytogenomic Nomenclature (ISCN, 2020, [41]).

### 4.4. DNA Analysis

Blood was obtained by venipuncture and collected in disposable plastic test tubes containing a preservative solution (0.5 M EDTA) at a ratio of 1:10 (preservative:blood). Genomic DNA was isolated from peripheral venous blood lymphocytes using the Wizard^®^ Genomic DNA Purification Kit (Promega Inc., Madison, WI, USA) according to the manufacturer’s protocol. 

Molecular analysis of CAG repeats was based on the method previously described in detail by Brown et al. (1989) [5]. The CAG*n* polymorphic locus of the *AR* gene was analyzed by PCR-based amplified fragment length polymorphism (AFLP-PCR or AFLP) method according to the approved and tested medical technology in the Research Centre for Medical Genetics (Moscow) using the MC2 thermocycler (DNA Technology, Moscow, Russia).

PCR was performed in a total volume of 25 µL, consisting of 11.25 µL of 2X PCR Master Mix (containing Taq polymerase, dNTPs, betaine and buffer), 0.25 µL of each forward and reverse primer (specific for the CAG repeat region), and 2.0 µL of template DNA. The primers used were designed to flank the CAG repeat region to facilitate amplification. The PCR conditions included an initial denaturation step at 95 °C for 2 min, followed by 33 cycles of denaturation at 94 °C for 30 s, annealing at 64 °C for 30 s, and extension at 72 °C for 30 s. A final extension step was performed at 72 °C for 3 min.

The amplified products were analyzed by polyacrylamide gel electrophoresis using the PowerPack Universal power supply (Bio-Rad Laboratories, Inc., Hercules, CA, USA). A 7% polyacrylamide gel was prepared, and the PCR products were mixed with loading dye before loading into the wells. λ DNA digested with *PstI* was used as a molecular weight marker. The gel was run at a constant voltage (typically around 150–200 V) until the dye front had migrated an appropriate distance, usually around 1–2 h depending on the gel concentration and voltage applied. 

After the separation of the fragments, the gel was stained with ethidium bromide (0.1 mg/mL in 1× TBE) for 20 min, washed with water, and photographed in transmitted UV light at a wavelength of 312 nm. A GEL DOC 2000 imaging station and the QUANTITY ONE^®^ software, version 4.6.8 (Bio-Rad Laboratories, Inc., Hercules, CA, USA) were used for analyzing and recording the resulting images. The lengths of the amplified fragments were determined by comparing their migration distance against a DNA ladder. The number of CAG repeats was calculated from the size of the PCR products observed on the gel. 

### 4.5. Statistical Analysis

Statistical data analysis was performed using various statistical methods (Pearson’s chi-squared test, and chi-squared test with the Yates correction) with the STATISTICA software version 10.0 (StatSoft Inc., Tulsa, OK, USA). The presence of significant differences between groups and subgroups was determined at a value of *p* < 0.05.

## 5. Conclusions

In this study, we analyzed the CAG*n* polymorphic locus of the *AR* gene in a large cohort of Russian men, comparing infertile individuals with fertile controls. The results showed a statistically significant difference in the frequencies of some CAG alleles between the two groups. An important point is the assessment of the frequency of the trinucleotide repeat expansion, which makes it possible to determine the prevalence of SBMA or its risk in Russian infertile men. The relatively high frequency of CAG repeat expansions in the *AR* gene in infertile male patients suggests the possibility of pre-symptomatic diagnosis of SBMA in these patients. This emphasizes the importance of analysis of the CAG*n* polymorphic locus of the *AR* gene in azoospermic and oligozoospermic men and referral for the analysis of CAG repeats in the *AR* gene to ensure the early detection and dynamic follow-up of individuals with trinucleotide repeat expansion.

## Figures and Tables

**Figure 1 ijms-25-12183-f001:**
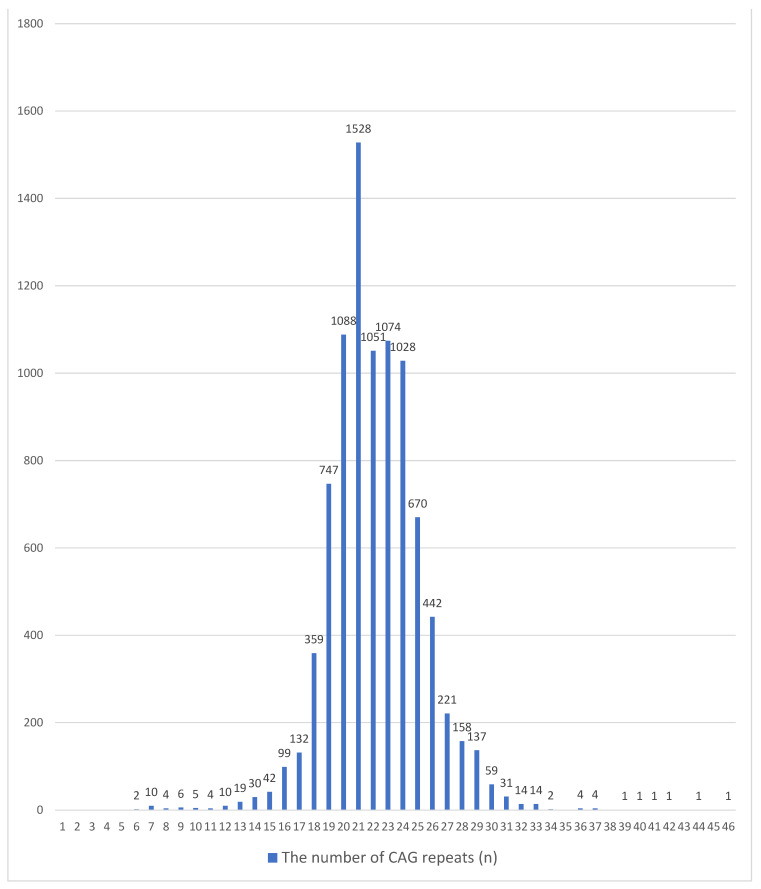
Distribution of CAG repeats in exon 1 of the *AR* gene in infertile Russian men (*n* = 9000).

**Figure 2 ijms-25-12183-f002:**
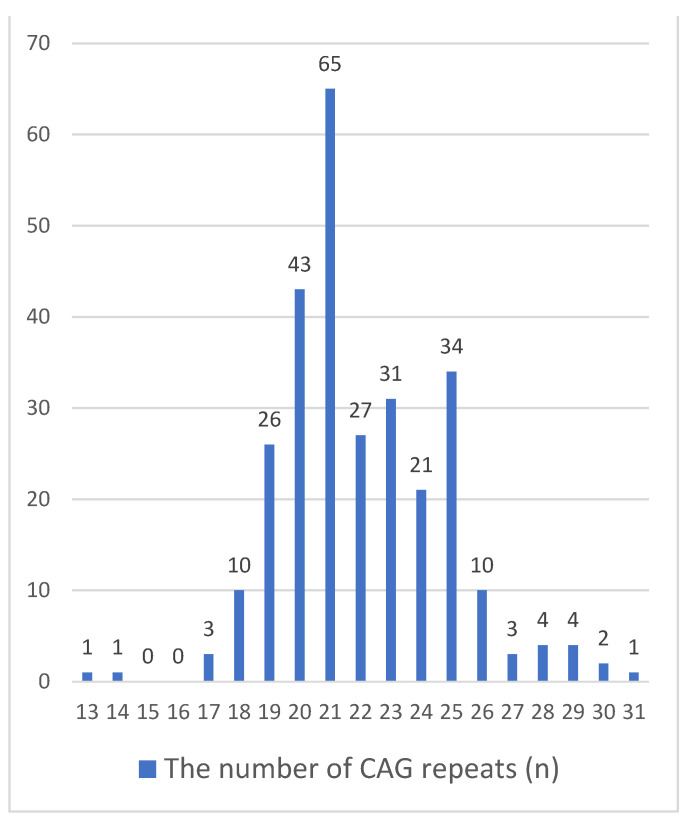
Distribution of CAG repeats in exon 1 of the *AR* gene in fertile Russian men (controls, *n* = 286).

**Table 1 ijms-25-12183-t001:** Characteristics of the CAG*n* polymorphic variants of the *AR* gene in fertile and infertile Russian men.

CAG Repeats	Infertile Men (*n* = 9000)	Fertile Men (*n* = 286)
Min-max, *n*	6–46	13–31
Mean ± SD, *n*	22.2 ± 3.1	22.0 ± 2.7
Mode, *n*	21	21
Median, *n*	22	21
Q25	20	20
Q75	24	24
«Short» alleles (≤18), *n* (%)	722 (8.02%) ^1^	15 (5.24%) ^1^
«Medium» alleles (19-25), *n* (%)	7186 (79.84%) ^2^	247 (86.36%) ^2^
«Long» alleles (≥26), *n* (%)	1092 (12.13%) ^3^	24 (8.39%) ^3^
CAG-repeats expansion	6	0

^1^ *χ*^2^ = 2.927; *p* = 0.088. ^2^ *χ*^2^ = 7.375; *p* = 0.007. ^3^ *χ*^2^ = 3.670; *p* = 0.056.

**Table 2 ijms-25-12183-t002:** Allele frequencies of polymorphic (CAG)*n* variants (*n* = 16–29) of the *AR* gene in Russian infertile and fertile men.

Variants of the CAG*n* * Polymorphic Locus of the *AR* Gene	Number of Patients, Allele Frequency (AF)		*p*-Value
	Russian Infertile Men (*n* = 9000)	Russian Fertile Men (*n* = 286)	
16	99 (0.0110)	-	-
17	132 (0.0147)	3 (0.0105)	0.742
18	359 (0.0399)	10 (0.0350)	0.675
19	747 (0.0830)	26 (0.0909)	0.634
20	1088 (0.1209)	43 (0.1450)	0.134
21	1528 (0.1698)	65 (0.2273)	**0.012**
22	1051 (0.1168)	27 (0.0944)	0.245
23	1074 (0.1193)	31 (0.1084)	0.574
24	1028 (0.1142)	21 (0.0734)	**0.032**
25	670 (0.0744)	34 (0.1189)	**0.006**
26	442 (0.0491)	10 (0.0350)	0.274
27	221 (0.0246)	3 (0.0105)	0.184
28	158 (0.0176)	4 (0.0140)	0.823
29	137 (0.0152)	4 (0.0140)	0.939

* Number of repeats. The *p*-values showing statistically significant differences between the groups are highlighted in bold.

## Data Availability

Data are contained within the article.

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
