# Peer review of "CAGn Polymorphic Locus of Androgen Receptor (AR) Gene in Russian Infertile and Fertile Men"

_ijms, 2024, doi:10.3390/ijms252212183_

Round 1

Reviewer 1 Report

Comments and Suggestions for Authors

General comments:

This study evaluated CAGn locus (variant) in a large cohort of infertile men and their frequency was compared with fertile men. In general, the study is well done and written regarding all the sections. The introduction well contextualizes the objective of the study, M&M and results are sufficiently reported. An extended discussion was made regarding appropriate literature citation and conclusion are supported by the results. Several limitations were reported to clarify the study design and helping the interpretation of the findings.  As mentioned by the authors, the small sample size of fertile men was one constraint for comparing data and contrast with the record of 9,000 infertile men. However, this is a retrospective study and can be not changed regarding this aspect. Finally, the authors suggest the possibility to include this genetic test for the evaluation in infertile patients in certain conditions in the discussion. Maybe a more explicit mention about this issue in the conclusion is adequate.  Some care is need regarding the punctuation (coma and points).

Specific comments:

L49: AR gene. Please confirm the use of “AR” abbreviation throughout the manuscript.

L183-189: There is a tendency for a higher frequency of "short" (n≤18) and "long" (n≥26) CAG alleles …; and yes, probably due the sample size of fertile men - size effect. But for "medium" (19-25) alleles, this is right. I suggest to rewritten this paragraph.

L191: Also add the reference in “Previously [24], …”.

L247: “…six patients…”.

L306: there is a name for this program?

Author Response

We would like to thank Reviewer for valuable and important comments. We have taken it into account and have revised the manuscript.

Point 1:  General comments:

This study evaluated CAGn locus (variant) in a large cohort of infertile men and their frequency was compared with fertile men. In general, the study is well done and written regarding all the sections. The introduction well contextualizes the objective of the study, M&M and results are sufficiently reported. An extended discussion was made regarding appropriate literature citation and conclusion are supported by the results. Several limitations were reported to clarify the study design and helping the interpretation of the findings.  As mentioned by the authors, the small sample size of fertile men was one constraint for comparing data and contrast with the record of 9,000 infertile men. However, this is a retrospective study and can be not changed regarding this aspect. Finally, the authors suggest the possibility to include this genetic test for the evaluation in infertile patients in certain conditions in the discussion. Maybe a more explicit mention about this issue in the conclusion is adequate.  Some care is need regarding the punctuation (coma and points).

Response: It has been corrected. We have corrected punctuation and typing errors throughout the manuscript, the Сonclusion has been supplemented.

Point 2: L49: AR gene. Please confirm the use of “AR” abbreviation throughout the manuscript.

Response: In order to ensure consistency, we have made the necessary corrections to the use of the abbreviation 'AR' throughout the manuscript.

Point 3: L183-189: There is a tendency for a higher frequency of "short" (n≤18) and "long" (n≥26) CAG alleles …; and yes, probably due the sample size of fertile men - size effect. But for "medium" (19-25) alleles, this is right. I suggest to rewritten this paragraph.

Response:  It has been corrected.

Point 4: L191: Also add the reference in “Previously [24], …”.

Response: It has been corrected.

Point 5: L247: “…six patients…”.

Response: It has been corrected.

Point 6: L306: there is a name for this program?

Response: There is no name of this program. The CAG polymorphic locus of the AR gene is one of genetic factors evaluated in this study.

Reviewer 2 Report

Comments and Suggestions for Authors

This paper explores the polymorphic variants of the CAGn locus responsible for the androgen receptor as an intriguing marker of male infertility as well as the potential of the technique to be used as a possible diagnostic tool for further pathologies.

I appreciate a large cohort of patients included in the study as well dedication by the authors to properly discuss the limitations of the study and possible future prospects which is quite interesting. Nevertheless, I do have several queries:

-              As disclosed by the authors, similar studies have been already done particularly on Eastern Slavic men (Russians, Ukrainians, etc.). The primary outcomes of previous reports should be disclosed in the Introduction section. This way the study can fortify its rationale and originality. I would strongly emphasize to outline particularly the originality of the paper, which currently relies on a large cohort of infertile patients. Major similarities in the experimental approach are found particularly in the Discussion section where previous reports are outlined in detail unraveling that a more complex approach was chosen to study different subpopulations of fertile/infertile subjects, offering more comprehensive data when compared to this paper. 

-              In this sense, given a very large cohort of patients, why was this group not divided into subpopulations of specific semen abnormalities (oligozoospemria, asthenozoospermia, teratozoospermia, etc.)?

-              What was the age of the subjects involved? (“reproductive age” is a very broad term in male reproduction)

-              The data collected from donors serving as fertile controls were already published as cited in the text (reference no. 24)? What exclusion and inclusion criteria were used in case of the donors?

-              I have several issues with the methodology, as a good portion of the chemicals are missing their manufacturers and almost no equipment is described in the text. The authors state that semen analysis was done according to the WHO manual, however a proper reference is missing. Which edition of the manual was used? How was concentration and motility evaluated – manually or by CASA? Please, briefly describe the generally accepted protocol for a standard cytogenetic study on FGA-stimulated cultures of peripheral blood lymphocytes or add a relevant reference. The same applies for the AFLP-PCR method.

Author Response

We would like to thank Reviewer for detailed evaluation of our manuscript. Your comments and recommendations are constructive and very helpful to improve the manuscript.

Point 1:  As disclosed by the authors, similar studies have been already done particularly on Eastern Slavic men (Russians, Ukrainians, etc.). The primary outcomes of previous reports should be disclosed in the Introduction section. This way the study can fortify its rationale and originality. I would strongly emphasize to outline particularly the originality of the paper, which currently relies on a large cohort of infertile patients. Major similarities in the experimental approach are found particularly in the Discussion section where previous reports are outlined in detail unraveling that a more complex approach was chosen to study different subpopulations of fertile/infertile subjects, offering more comprehensive data when compared to this paper.

Response: 

This remark was taken into account. The Introduction section has been supplemented with information about the studies done earlier on Russian and Ukrainian men samples and their brief analysis.

Point 2:  In this sense, given a very large cohort of patients, why was this group not divided into subpopulations of specific semen abnormalities (oligozoospermia, asthenozoospermia, teratozoospermia, etc.)?

Response: 

Because we did not have ejaculate samples from many patients, we were not able to perform semen analysis on all study participants and stratify the group of infertile men into subgroups according to semen parameters and abnormalities. This analysis was not part of the study objectives, although it would be very interesting. Thank you for your comments, which we will take into account in future research.

Point 3:  What was the age of the subjects involved? (“reproductive age” is a very broad term in male reproduction)

Response:  Both fertile and infertile men enrolled in the study were mostly between the ages of twenty and forty. In general, their ages were similar. As there was no precise information on the age of all the people in the study, the average age of the groups is not given.

Point 4:  The data collected from donors serving as fertile controls were already published as cited in the text (reference no. 24)? What exclusion and inclusion criteria were used in case of the donors?

Response: The control group consisted of 286 unrelated fertile Russian men aged ≥ 18 years. All men in the control group were the father of one or more children born from a natural pregnancy. All were residents of the Russian Federation, predominantly ethnic Russians to maintain genetic homogeneity. Other inclusion criteria required that participants had no history of male infertility and serious chronic diseases, and that paternity was confirmed by DNA analysis. Exclusion criteria were male infertility, known genetic disorders that could affect fertility, and those who did not consent to DNA testing. These criteria clearly define the study control group and ensure that the results are relevant to understanding the genetic factors associated with fertility in this population.

Point 5:  I have several issues with the methodology, as a good portion of the chemicals are missing their manufacturers and almost no equipment is described in the text. The authors state that semen analysis was done according to the WHO manual, however a proper reference is missing. Which edition of the manual was used? How was concentration and motility evaluated – manually or by CASA? Please, briefly describe the generally accepted protocol for a standard cytogenetic study on FGA-stimulated cultures of peripheral blood lymphocytes or add a relevant reference. The same applies for the AFLP-PCR method.

Response: It was corrected. Sperm concentration, motility and morphology were assessed manually according to WHO recommendations, 5th edition. This information has been added to the Materials and methods.
